# Effective Approaches to Study the Genetic Variability of SARS-CoV-2

**DOI:** 10.3390/v14091855

**Published:** 2022-08-24

**Authors:** Ivan Kotov, Valeriia Saenko, Nadezhda Borisova, Anton Kolesnikov, Larisa Kondrasheva, Elena Tivanova, Kamil Khafizov, Vasily Akimkin

**Affiliations:** 1FSBI Central Research Institute for Epidemiology of the Federal Service for Surveillance of Consumer Rights Protection and Human Wellbeing, 111123 Moscow, Russia; 2Moscow Institute of Physics and Technology, National Research University, 115184 Dolgoprudny, Russia

**Keywords:** coronavirus, SARS-CoV-2, NGS, sequencing, bioinformatics, multiplex PCR

## Abstract

Significant efforts are being made in many countries around the world to respond to the COVID-19 pandemic by developing diagnostic reagent kits, identifying infected people, determining treatment methods, and finally producing effective vaccines. However, novel coronavirus variants may potentially reduce the effectiveness of all these efforts, demonstrating increased transmissibility and abated response to therapy or vaccines, as well as the possibility of false negative results in diagnostic procedures based on nucleic acid amplification methods. Since the end of 2020, several variants of concern have been discovered around the world. When information about a new, potentially more dangerous strain of pathogen appears, it is crucial to determine the moment of its emergence in a region. Eventually, that permits taking timely measures and minimizing new risks associated with the spreading of the virus. Therefore, numerous nations have made tremendous efforts to identify and trace these virus variants, which necessitates serious technological processes to sequence a large number of viral genomes. Here, we report on our experience as one of the primary laboratories involved in monitoring SARS-CoV-2 variants in Russia. We discuss the various approaches used, describe effective protocols, and outline a potential technique combining several methods to increase the ability to trace genetic variants while minimizing financial and labor costs.

## 1. Introduction

The novel coronavirus infection (COVID-19), caused by a virus named Severe Acute Respiratory Syndrome Coronavirus 2 (SARS-CoV-2), was first reported in late 2019 in Wuhan, China, and then quickly spread around the world [1,2]. After 2.5 months, on 11 March 2020, the World Health Organization (WHO) officially announced the start of the COVID-19 pandemic. Despite numerous and lengthy quarantines around the world, there were over 580 million confirmed cases of COVID-19 at the end of July 2022, including nearly 6.4 million deaths (https://www.worldometers.info/coronavirus/, accessed on 22 December 2021). Notably, since the end of 2020, the international scientific community has described several suspicious variants of SARS-CoV-2 that require special attention. These primarily include Alpha (B.1.1.7), Beta (B.1.351), Gamma (P.1), Delta (B.1.617.2), and the highly transmissible variant Omicron (B.1.1.529), including its many sub-lineages. These virus strains attracted the interest of researchers after reports of an increase in human-to-human transmission began to appear in a number of geographic locations, and afterward, new variants of the pathogen were discovered all over the world. The rapid spread of some strains has shown that re-infections are possible even in previously infected and vaccinated people [3,4,5], despite being noted that such re-infections are often mild or even asymptomatic [6,7,8]. In addition, new mutations may affect the performance of diagnostic assays based on nucleic acid amplification [9,10]. Thus, identifying new variants of SARS-CoV-2 and tracking their spread around the world plays an important role in the fight against the pandemic.

Based on the effects on transmissibility, mortality, and response to therapy, the WHO has created a variant classification scheme that divides SARS-CoV-2 variations into four categories: Variants of Concern (VOC), Variants of Interest (VOI), Variants Under Monitoring (VUM), and Formerly Monitored Variants (FMV) (https://www.who.int/activities/tracking-SARS-CoV-2-variants, accessed on 22 December 2021). Variants, designated as VOC, have a number of proven traits, such as increased transmission ability, more severe disease (an increase in the number of hospitalizations or deaths), change in the disease course, and a significant decrease in neutralization by antibodies formed in response to a previous infection or vaccination, all of which can reduce the effectiveness of treatment and preventive measures. Currently, the VOC group is represented by the Omicron (B.1.1.529) subtype and all their descendants, including Omicron sub-lineages BA.1-5 and several recombinant forms such as XE. The VOI category includes variants with specific genetic markers that have been associated with changes to receptor binding, reduced neutralization by antibodies, reduced efficacy of treatments, potential diagnostic impact, or predicted increase in transmissibility or disease severity. Furthermore, the impact of VOI variants on the global epidemiological situation has been proven, for example, an increase in relative prevalence is observed in various nations. There are currently no strains in this category as of July 2022. Previously, this category included Epsilon (B.1.427, B.1.429), Lambda (C.37), Mu (B.1.621), and others. The variants in the VUM category have genetic changes that are expected to influence the pathogen’s properties or may affect them in the future, but there is no clear evidence of this currently. To date, there are no lineages labeled as VUM. The variants labeled as FMV used to belong to other presented classes, but were reclassified later due to dropping their proportions to zero or near-zero levels over time or lack of impact on the epidemiological situation. Furthermore, the strain can be assigned to this class in the light of new data indicating the absence of new properties in the variant. The FMV category includes lineages AT.1, B.1.1.523, and many others with all of their descendants.

Variants of SARS-CoV-2 are characterized by changes in their nucleotide sequence, which can be identified in a variety of ways. Undoubtedly, the most informative method is whole genome sequencing, which allows us to collect and analyze the maximum amount of information about genomic changes. However, this approach is quite expensive and can be time-consuming, and new methods for faster and cheaper genotyping are also under development to overcome these limitations.

Numerous protocols based on RT-PCR and LAMP methods with variant-specific primers have already been developed, allowing reliable identification of the virus strain based on the results of several reactions [11,12,13,14,15,16,17]. Unfortunately, these relatively cheap and fast methods are not universal for detecting different virus variants. Primers and/or fluorescent probes must be designed for each discrete mutation whose presence has to be verified, which considerably slows down and complicates the process of identifying new variants. In addition, genetic changes emerging in the primer annealing sites can significantly disrupt the amplification efficiency, which is especially relevant for loop-mediated isothermal amplification since this technology requires a longer total primer annealing area than PCR assays. Finally, these methods are unsuitable for the new virus variant detection and can only be used when the new strain and its features have already been described using NGS sequencing.

Sequencing of part or the whole genome remains the primary way to confirm the presence of known mutations and identify the new ones. Determining the S-protein gene sequence may be sufficient to detect most of the known strains since the majority of them are characterized by the presence of specific mutations in this important genome region. Nevertheless, sequencing the entire viral genome by Next-Generation Sequencing (NGS) techniques is the most exact method for studying the pathogen, allowing identification and detection of all possible variants of the SARS-CoV-2 sequence. However, despite the small size of viral genomes (∼30,000 bp for SARS-CoV-2) and the recent decline in the cost of sequencing technologies, this method still requires significant material, time, and labor resources. This fact does not always allow the use of NGS in routine practice, especially in conditions of a large sample flow. In addition, not all laboratories have the ability to carry out large-scale high-throughput sequencing experiments.

Among the popular approaches for sequencing the SARS-CoV-2 genome, in addition to a large number of available commercial solutions, the enumeration of which is beyond the scope of this work, the ARTIC primer set (https://artic.network, accessed on 22 December 2021). and its several modifications are widely used. That allows amplification of the entire viral genome during multiplex PCR, followed by preparing libraries for sequencing on Illumina or Oxford Nanopore Technologies instruments. The advantage of such an approach is the ability to detect new mutations and pathogen variants in addition to tracking those already known. However, this method can be rather time-consuming [18,19,20,21,22,23].

In this paper, we describe our experience in detecting and identifying SARS-CoV-2 variants using approaches developed in our laboratory to present efficient and low-cost methods suitable to detect and estimate the prevalence of SARS-CoV-2 strains. For quick genotyping of viruses using NGS technologies, in some cases, it is advisable to use S-protein gene sequencing with the modifications proposed, which helps to reduce the experiment costs. For the more detailed studying of the pathogen, the whole genome sequencing panel may be helpful, in which we have made some changes that reduce financial costs.

## 2. Materials and Methods

For sequencing, nasopharyngeal swabs were used from patients with symptoms of coronavirus infection, for whom the presence of SARS-CoV-2 was confirmed by real-time PCR using the AmpliSens^®^ COVID-19-FL reagent kit (AmpliSens, Moscow, Russia). The samples were placed in a transport medium. Isolation of RNA from clinical material was performed using the RIBO-prep reagent kit (AmpliSens, Russia). Mainly those clinical samples were selected in which the value of the cycle threshold (Ct) during PCR did not exceed 25. The reverse transcription reaction was performed using 10 μL of RNA samples using the Reverta-L kit (AmpliSens, Russia) according to the manufacturer’s instructions. The obtained cDNA was used as a template for the amplification of genome fragments.

### 2.1. S-Protein Gene Sequencing

For sequencing the entire S-protein gene, we employed amplification with a panel of primers containing Illumina adapter tails. Multiplex PCR amplification was performed in a volume of 25 μL in four separate reactions (see Appendix A; the last version of primers for whole S-protein gene sequencing is presented in Appendix A) containing 5 μL of template cDNA, using 10 μL of PCR mix-2 blue (AmpliSense, Russia) containing Taq polymerase, 2.5 μL dNTP 4.4 mM (AmpliSense, Russia), primer pools 0.5 μL (final concentration of each primer in the reaction mixture 0.1 pmol/μL), and sterile water 7 μL. Amplification profile: (1) denaturation at 95 °C for 2 min; (2) 39 amplification cycles: 95 °C—30 s, 60 °C—20 s, 72 °C—60 s; (3) final elongation at 72 °C for 3 min. Next, PCR products were visualized using electrophoresis in 1.7% agarose gel stained with ethidium bromide. The amplified fragments were mixed according to the visual assessment of the concentrations in the ratio 5a:6b:7c:2d, where a, b, c, and d are the numerical values of the assessments for each pool. Pooled PCR products were purified using AMPure XP beads (Beckman Coulter, Indianapolis, IN, USA) at a ratio of 1:1.1. The library preparation process was carried out using the same polymerase; normally 7–10 amplification cycles were carried out. Purified fragment concentrations were measured using the Qubit dsDNA HS Assay Kit with a Qubit 4.0 fluorimeter (Invitrogen, Waltham, MA, USA). High-throughput sequencing was performed on the Illumina MiSeq platform with the MiSeq Reagent Kit v2 (300 and 500 cycles) and v3 (600 cycles).

### 2.2. Whole Virus Genome Sequencing

We used a tailed amplicon approach for whole genome sequencing of SARS-CoV-2. Viral fragments were amplified using modified ARTIC (v3) primers containing Illumina adapter tails. The multiplex PCR amplification reaction was performed in a total reaction volume of 25 μL in five separate reactions (see Appendix A), containing 5 μL of template cDNA, using 10 μL of PCR mixture-2 blue (AmpliSense, Russia) with Taq polymerase, 1.8 μL dNTP 4.4 mM (AmpliSense, Russia), primer pools 0.15 μL, and sterile water 8.05 μL. PCR conditions were the following: initial denaturation at 95 °C for 2 min, followed by 40 cycles of 95 °C for 15 s, annealing temperature at 63 °C for 30 s, 72 °C for 1 min 20 s, and final extension at 72 °C for 3 min. Subsequently, PCR products were visualized using electrophoresis in 1.7% agarose gel stained with ethidium bromide. The amplified fragments were mixed according to a visual assessment of concentrations. PCR products were purified using AMPure XP beads (Beckman Coulter, Indianapolis, IN, USA) at a ratio of 1:1x. The library preparation process was performed using the same Taq polymerase; normally 10–12 amplification cycles were carried out. Purified fragment concentrations were measured using the Qubit dsDNA HS Assay Kit with a Qubit 4.0 fluorometer (Invitrogen, Waltham, MA, USA). High-throughput sequencing was performed on the Illumina MiSeq platform with the MiSeq Reagent Kit v3 (600 cycles). The number of samples for loading is up to 100 libraries on MiSeq v3 (600 cycles). This coverage is normally sufficient to obtain the whole genome sequence of SARS-CoV-2.

### 2.3. Bioinformatics Analysis

First of all, the sequencing reads were quality controlled. The adapter sequences were removed from reads using the bbtools package [24], and then all reads shorter than 70 bases or with an average quality of less than 20 were filtered out. Bases with a quality of less than 20 were also removed from the right end of each read. Second, the bwa utility was used to map prepared reads onto the reference sequence. Using the samtools package [25], the resulting sam files were converted to bam and sorted. The obtained data were used to create consensus fasta sequences with custom Python scripts. A region of the viral genome was considered successfully read if it was covered by at least ten reads. The acquired sequences were aligned to the reference genome using MAFFT [26] to ease the comparative analysis. The Nextclade [27] tool was used to identify the strains present in the studied samples.

## 3. Results

### 3.1. S-Protein Gene Sequencing

Previously, we developed a primer panel consisting of five pairs of oligonucleotides. This allowed us to perform targeted amplification of the SARS-CoV-2 genome regions containing important mutations in the S-protein gene (including amino acid substitutions K417T, L452R, T478K, E484K, S494P, N501Y, A570D, P681H, etc., and deletions of HV69-70 and Y144), followed by preparation for sequencing. The primers were modified with Illumina adapter tails included at the synthesis stage, making it possible to carry out the indexing stage immediately after amplification [28]. The simplified protocol dramatically reduces the cost of sample preparation, and increases the number of samples analyzed in a single sequencing run. As an example of applying this approach, we demonstrated the prevalence of virus variants in the Moscow region in the period from February to June 2021, which was marked by a change in the dominant strains, with the Delta almost completely replacing all other variants by July 2021, as further studies showed [29].

Although the approach described above helped to quickly genotype virus samples and uniquely assign them to a definite strain, even at that time, it became clear that new variants with different mutations would continue to appear in the future. Therefore, there is a high chance that they will appear in the S-protein gene fragments uncovered by the panel. It was subsequently confirmed with the appearance of subtype AY.4.2. It is one of the Delta variant sub-lineages, which received the name “VUI-21OCT-1” in October 2021 and was designated as a VUI. Finally, the emergence of the Omicron strain with a very high number of mutations in the genome, especially in the S-protein gene [30], clearly demonstrated the importance of determining the sequence of the entire gene. In this regard, we have developed a new panel for targeted amplification of the S-protein gene, which allows for effective detection of all its genomic changes.

The primers for targeted amplification of the S-protein gene were selected manually, taking into account the available information on conservative regions of the genome. The estimation of primer melting points and interactions between them were performed using the Multiple Primer Analyzer tool (ThermoFisher Scientific, Waltham, MA, USA). Using the BLASTn [31] program, the specificity of each sequence obtained for all known organisms (primarily human, whose genetic material is present in the sample in the largest amount) was evaluated. This approach helps exclude many nonspecific interactions of the primers with human and other organisms’ genomes. The resulting 20 primer pairs were modified with Illumina adapter tails. This modification speeds up the process of obtaining libraries for targeted sequencing—after PCR enrichment of target cDNA fragments in four independent reactions, the stage of dual indexing immediately begins. The length of amplicons was made to be ∼300 bp, allowing for sequencing the resulting libraries on the Illumina platform using various reagent kits: MiSeq v2 (300 cycles), v2 (500 cycles), and v3 (600 cycles). The amount of samples for loading is up to 356 libraries on Miseq v2 (300 cycles) and up to 468 for v2 and v3 (500 and 600 cycles, respectively). This value is limited by the number of indexes. Therefore, using the custom ones, it is potentially possible to expand the number of libraries. Most often, the number of samples in one run fluctuated around 300. The coverage of individual regions can reach from almost zero values to tens of thousands of reads. However, even if some regions do not have very high coverage, but all are covered enough, this is the favorable outcome. Furthermore, a separate region was deemed successfully sequenced if it was covered by at least 10 reads. The timeline of coverage quality is shown in Figure 1. Even though the width of the interval is highly dependent on the quality of the supplied samples, it can be utilized to evaluate the efficacy of amplification before sequencing.

As can be seen from the timeline, there are 95% confidence region dramatic narrowings at high-value areas in July and December of 2021. In time, they correspond to replacements of low coverage primer pairs with more effective ones. Then, when new sublines spread, the 95% confidence region gradually widens, and the average numbers of sufficiently covered regions fall, indicating degradation in resulting data quality. This circumstance necessitates an update to the primer set for the amplification of the mutated genomes. Furthermore, the proportion of samples having 10 or more regions covered by less than 10 reads rose to over 2.5% within roughly 4 months. The case described corresponds to a dropping of the lower boundary of the 95% confidence region below the level of the worst acceptable coverage threshold of the S-protein gene. This notable observation enables us to calculate the typical primer set obsolescence time for use with the virus under investigation and to make other deductions on the virus’s evolution rate.

Initially, some primer pairs provided near-zero coverage in a significant percentage of cases, although it varied from sample to sample. Moreover, the widespread adoption of the Delta variant worsened the situation. After comparing the effectiveness of the primers and performing in silico research, it was discovered that the formation of unfavorable secondary structures in the reaction mixture, such as primer’s homo- and heterodimers, as well as mutation presence in the regions of primer–matrix complex formation, were the culprits of the problem. In order to solve the mentioned issues, the primers can be modified or replaced. However, when it is necessary to change several primers that are part of a large pool, this procedure becomes very laborious since each candidate oligonucleotide must be verified for the dimer formation with a large number of rest primers that make up the pool. Moreover, this procedure is usually performed iteratively, i.e., each new primer candidate is compared with existing ones, without understanding what challenge it can cause for primers will be designed next. Therefore, the objective of finding the optimal set becomes rather complicated. To facilitate the updating and improvement of large primer pools, we developed a software pipeline that allows for partial automation and, as a result, acceleration of the process of designing new primers for large panels (Figure 2).

If it is necessary to replace the primer in the pool, the selected sequence of the genome region is given as input to the software pipeline, within which the annealing site of the new primer should be located. Usually, a neighborhood of the old primer annealing site is chosen. The boundaries of the input region are largely determined by the requirements for the length of the resulting amplicon. From the string received as an input, the pipeline generates a list of all possible k-mers in a given range of lengths; by default, k takes values from 18 to 30. If a reverse primer option is set, then k-mers are made based on a reverse complementary strand to the original. The resulting sequences are then filtered by the melting point of the primer–template complex ($T_{m}$), which is estimated with the technique used in the Multiple Primer Analyzer from Thermo Fisher. Unless otherwise specified, at this stage, from the previously generated k-mers, those that have $T_{m}$ in the range of 62–64 °C are selected. If necessary, the selected oligonucleotides can be modified with adapter sequences: Nextera adapter sequences (Nextera PE adapter sequences) were used in the present study.

Next, the resulting adapter–primer structures are tested for the formation of homodimers. Oligonucleotides from the list of those that passed these checks go through the procedure of “fitting” to the initial primer pool one by one. For each one, the dimer formation metric with all pool primers is calculated. The value of the Gibbs free energy change (ΔG) of dimers formed by the candidate oligonucleotide with pool primers was used as such a metric. After the metric values are obtained for all sequences, the set of candidates is sorted by its value. Next, the researcher has only to find the most suitable oligonucleotides at the top of this list. The advantage of this approach is that it can suggest several good candidates, which can be helpful in many cases. For example, if the best sequence, which does not form dimers with pool primers, for some reason, does not suit the researcher, e.g., it may have a frequent mutation at its annealing site, which prevents amplification. Such situations often occur in practice, and to avoid them, it is preferable to prepare three or more oligonucleotides at once. Notably, the quality of the new primers selection depends on the accuracy of the instruments that detect the formation of dimers and hairpins. In our implementation of the pipeline used in this study, the PrimerDimer (http://primer-dimer.com, accessed on 22 December 2021) was used as such a tool. On the other hand, other similar programs could hypothetically be employed as well.

Thus, in this work, we have developed a pipeline that allows for a quick replacement of individual primers in large pools if they form dimers, possess too high or too low melting temperatures, or have lost their effectiveness due to mutations in their annealing sites. If necessary, this tool can be used multiple times to replace a number of primers from the same pool to improve the overall performance of the set. With a large pool of primers, it is not always possible to achieve a complete absence of dimers, but even in such cases, the pipeline helps to obtain a pool with a minimum number of undesirable structures. An example of the above-mentioned tool usage is presented in Figure 3. Initially, there were three regions (7, 9 and 14) with very unstable coverage, which was apparently caused by the poor performance of the respective primers. After the discovery of low-quality primers, they were replaced with new ones according to the described scheme, which finally helped to obtain the desired coverage profile. Although some regions (such as 1 and 20) showed low coverage (median coverage rates are approximately 14 and 68 reads, respectively), the coverage rates of all regions of the S-protein gene became more stable and uniform, which is critical for obtaining a good consensus sequence.

### 3.2. Whole Virus Genome Sequencing

For the whole genome sequencing of SARS-CoV-2, a multitude of different approaches can be used, including the application of various commercially available primer panels for amplifying the virus genome (Swift Amplicon™ SARS-CoV-2 Panel; QIAseq SARS-CoV-2 Primer Panel; CleanPlex^®^ SARS-CoV-2 FLEX, etc.) However, because most commercial reagent kits are expensive and can also become obsolete, researchers have worked to create effective whole genome sequencing methods. One of these protocols is the ARTIC primer panel, which was created by an international consortium. This panel has undergone numerous modifications during the COVID-19 pandemic, the latest version being v4.1. However, the ARTIC approach requires a lot of laboratory time and is not always suitable for rapid preparation for whole genome sequencing. We decided to change strategy and use Illumina adapter tails along with ARTIC (v3) primer sequences. Bypassing costly and time-consuming methods, this approach enables the synthesis of libraries in two phases: (1) viral cDNA amplification and (2) double barcoding (indexing) libraries. It is well known that primers can become obsolete relatively quickly—single nucleotide mismatches at the annealing sites greatly slow down the amplification of the target fragments or even make it impossible, for example, in cases where the mutation occurs at the 3′-end of the primer. Because of this, we modified several primers that did not adequately cover some regions due to new mutations and changing the dominant variant circulating in the country. The original ARTIC allows multiplex amplification in two reactions, but with the addition of adaptor sequences, a significant number of uncovered genome fragments were obtained as a result. Thus, even after changing the primers’ structures, we encountered the dimer formation issue again. The problem is that during amplification of several genome fragments in one pool, primers from different pairs occasionally form dimers, which can also inhibit the copying of the target DNA fragment. Initially, we made an attempt to manually separate primer pairs and increased the number of pools, as proposed in the article [32]. However, this approach did not completely reach our goal because some parts of the genome were still not regularly covered. Thus, quantitative analysis of primer sequences in silico can be beneficial for evaluating secondary structures and the formation of various dimers.

Taking into account all of the above, we developed a genetic-like algorithm, based on the analysis of ΔG, to determine how to divide primer pairs into the fixed number of pools the most effective way. Figure 4A illustrates this algorithm’s fundamental concept. First, all primers are separated into five pools randomly. Next, the “fitting” process is carried out for each pair. The chosen pair is sequentially added to each pool, the fitness function values are calculated, and the pair is then added to the most suitable pool in a way that optimizes the fitness function. As a fitness function, the total ΔG of dimers formed in all pools (should be maximized) or the sum of squares ΔG of primer dimers in all pools (should be minimized) can be selected. For the total ΔG estimation, PrimerDimer web-based tool was also used [33]. However, minor adjustments in the script allow using any other tool to determine ΔG of dimers.

When using the described algorithm with the ΔG minimization option, the total ΔG changed with each optimization step, as shown in Figure 4B. For comparison, the red line shows the result previously achieved manually. After approximately 275 optimization steps, the total ΔG reached a plateau. It should be noted that the optimum achieved with the proposed algorithm is local. Based on the described principles, new primer distribution into pools was acquired, and these pools were then mixed and used to amplify new samples. Following that, samples were prepared in both old and new pools and then sequenced in one run. The results obtained are shown in Figure 5. Although this approach did not completely solve the low coverage problem, it increased both qualitatively and quantitatively.

Taking into account the nature of the distribution of examined samples by the coverage, the Mann–Whitney U test was used to statistically evaluate the impact of the new pooling. *p*-values < 0.05 were considered significant. According to its results, 56 pairs began to perform better, including eight pairs whose coverage increased from near-zero to 10 or more reads. For the other two pairs, the positive effect of the new pooling was not enough to reach coverage of at least 10 reads. The performance of 21 primer pairs deteriorated after pooling. However, 19 of them still managed to maintain the amplification efficiency at the level of 10+ reads. For example, a pair that used to give thousands of reads now provides hundreds. Another pair that used to provide adequate coverage no longer worked. Additionally, one primer pair began to work worse, which previously did not yield a 10+ read coverage. For the remaining 21 pairs, no significant changes were observed. Five of them failed to achieve the required minimal coverage in both pool configurations. Thus, if in the previous pools 82 primer pairs provided reliable coverage, then in the new pools, 89 pairs began to provide it. Further optimization of the primer set will thus be significantly accelerated and much more cost-effective. In addition, the amplification efficiency has been increased for a number of well-performing previous pairs, which may allow their concentration to be reduced and overall utilization to be improved.

## 4. Discussion

The spread of SARS-CoV-2 variants around the world has highlighted the need to develop specific tools aimed at detecting and monitoring their transmission in the population. Different approaches can be used to trace a known strain [34], including S-protein gene sequencing [28], but still, the most detailed method to identify new variants is sequencing the whole viral genome since it has already been reported that mutations outside the S-protein gene can change the infectivity of the virus, and even the prognosis of the disease [35,36,37,38]. In addition, the information about the whole pathogen genome allows scientists to obtain maximum information, along the way allowing answering some questions, such as when the imports into the country were happening, the evolution and distribution of variants in the territory, etc. [39].

In this work, we proposed cost-effective amplicon-based methods for sequencing SARS-CoV-2 genomes, which can be used for complete S-gene and whole genome sequencing. The last version of primer sets for both S-protein gene and whole genome sequencing are presented in the Appendix A. It should be noted that in the case of a large number of samples, the cost of NGS sequencing drops sharply and becomes even cheaper than Sanger sequencing. In such cases, the use of the latter method may be limited to the confirmation of mutations in the S-protein gene in the analysis of a limited number of samples [40].

In the course of evolution, the virus genome is constantly changing, and new mutations lead to a deterioration in the amplification of genome fragments, and, accordingly, the results of NGS sequencing. This is especially true for the S-protein gene, in which mutations occur most frequently. It is known that many of them can change the properties of the virus, making it more transmissible and avoiding the neutralizing effect of antibodies caused by vaccination or previous COVID-19 disease. However, using the algorithms developed in this work, we have made it possible to quickly optimize primer pools and modify the structures of separate oligonucleotides used to amplify the pathogen genome and again obtain high-quality results at a relatively low cost. All the scripts that used primer pool optimization in this paper are presented on github (https://github.com/Samurnihs/multiplexPCRHelp, accessed on 22 December 2021). The approaches developed were actively used by us throughout 2021–2022, and the sequencing results were uploaded to the national database VGARus (Virus Genome Aggregator of Russia), and GISAID [41]. Based on such genomic data, it is possible not only to monitor the effectiveness of diagnostic tests based on nucleic acid amplification methods and preventive vaccines, but also to predict the next rises in the incidence and the effectiveness of the introduction of anti-epidemic measures.

At the same time, it should be noted that such work is usually carried out by laboratories with good resources and qualified personnel. When sequencing is not available, as in small diagnostic laboratories, a good solution may be to link with other centers capable of performing NGS analysis on a large number of samples, which is the case of our institute. In the context of the need to identify and track variants of SARS-CoV-2 in a given area, this type of organization that monitors the genetic variability of the pathogen can perform NGS sequencing.

## Figures and Tables

**Figure 1 viruses-14-01855-f001:**
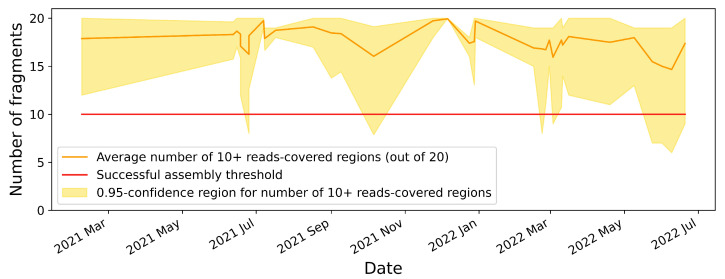
Timeline of coverage quality for S-protein gene sequencing from March of 2021 to July of 2022. The orange line represents the average number of 10+ read-covered regions (out of 20) in a single run. Yellow region shows 95%-confidence region for runs (2.5% of the best and worst results are discarded). The red line demonstrates the level of 10 regions covered by at least 10 reads, which is already a sample sequencing failure.

**Figure 2 viruses-14-01855-f002:**
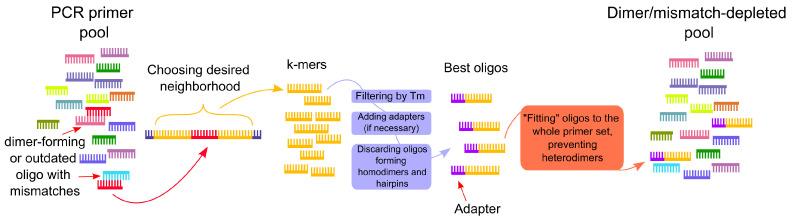
Schematic diagram of the pipeline for updating primer pools.

**Figure 3 viruses-14-01855-f003:**
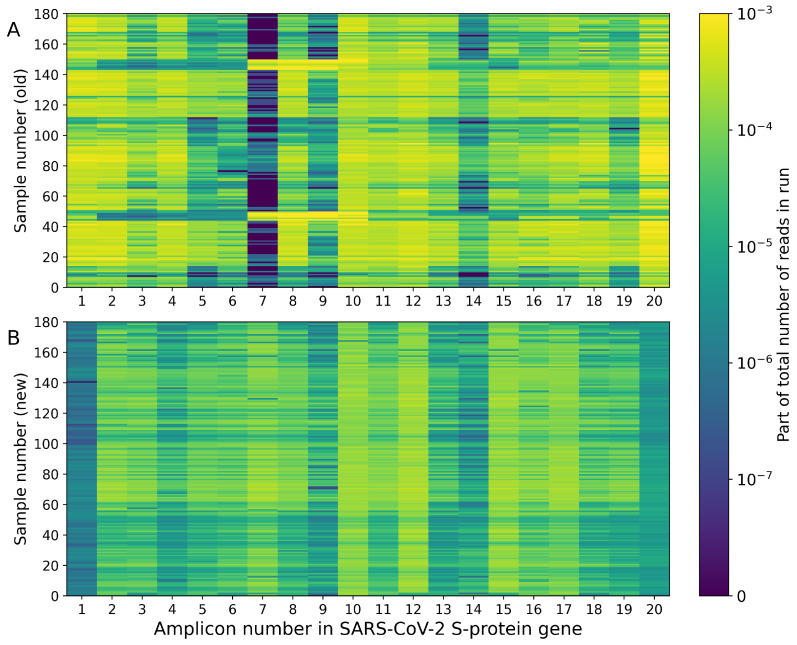
Coverage rates for 20 target regions of SARS-CoV-2 S-protein gene. Two series of 180 samples are presented. The upper color mesh (**A**) shows the coverage drop after the Delta variant became widespread. Several primer pairs became unstable-working, especially pair 7, which demonstrated zero coverage in the majority of cases. After the primer replacement, lower color mesh (**B**) was obtained. More uniform coverage allowed for an increase in the number of samples per sequencing run, and also made it possible to obtain high-quality consensus sequences.

**Figure 4 viruses-14-01855-f004:**
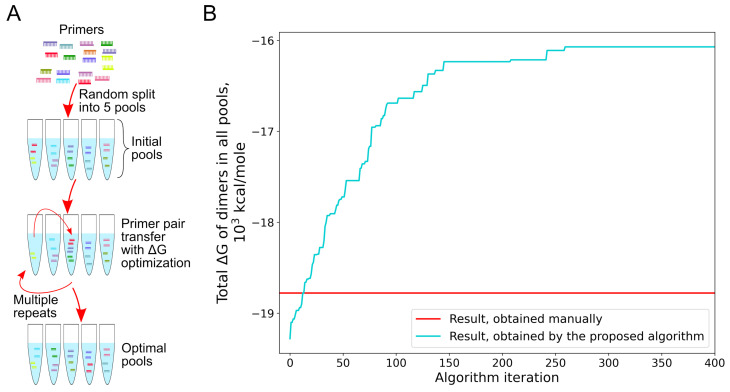
(**A**) A scheme of the iterative algorithm for optimizing large primer pools. (**B**) The graph shows the relationship between the number of optimization steps and the total ΔG of all generated dimers across all pools. One optimization step means “fitting” procedure for a single primer pair with its possible transfer between pools.

**Figure 5 viruses-14-01855-f005:**
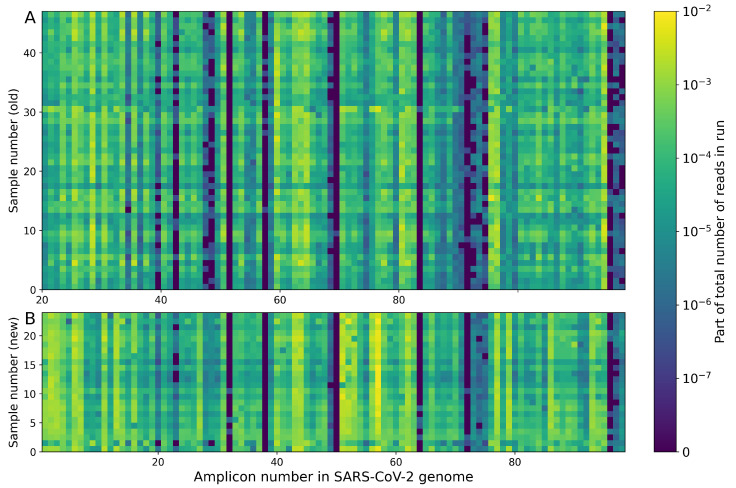
Coverage rates for 98 target regions of SARS-CoV-2 whole genome. Two series of 47 and 24 samples are presented for old (**A**) and new (**B**) poolings, respectively.

## Data Availability

All the sequences obtained in this work are published in the VGARus database. It can be accessed upon request.

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
