# Peer review of "Effective Approaches to Study the Genetic Variability of SARS-CoV-2"

_viruses, 2022, doi:10.3390/v14091855_

Round 1

Reviewer 1 Report

In this manuscript, the authors described a pipeline to design efficient primer pools for amplification and library preparation, and then detect the spike or whole genome sequences of SARS-CoV-2 by Illumina sequencing. The study is technically correct and sound. However, several concerns must be address before further consideration of this manuscript for publication.

Major points:

1. The authors declared their methods as “cost-effective”, so a detailed evaluation of the overall cost of their pipeline, from primer synthesis to sequencing cost, is needed. Comparison with other popular methods is also necessary.

2. Only Delta samples are analyzed in the study. However, the authors highlighted that the described pipeline could quickly optimize the primers to fit emerging variants, and this declaration should be verified.

Minor points:

1. Line 51. “VOI variants” should be VOIs, as “VOI” includes the term “variant”.

2. Line 69. The full name of “RT-PCR” and “LAMP” should be provided at their first appearance.

3. Line 80. “NGS sequencing” should be “next-generation sequencing (NGS)”, as “NGS” includes the term “sequencing”. And Line 85, just use “NGS” instead.

4. Figure 5. The x-axis labels are different in Fig.5A/B (20~120 for A and 0~100 for B), and should be explained.

Author Response

First of all, we would like to express our deep gratitude for the careful evaluation of our manuscript and insightful comments. To thoroughly address all the Reviewer’s remarks, we have carefully revised the manuscript. Please find below the point-by-point description of the revisions according to your comments.

1. The authors declared their methods as “cost-effective”, so a detailed evaluation of the overall cost of their pipeline, from primer synthesis to sequencing cost, is needed. Comparison with other popular methods is also
necessary.

This remark on your part is entirely accurate. Due to variable prices for reagents in our country, we roughly estimated that sequencing of a single sample became 40-50% less expensive. Lines 378–388 of the updated version of our manuscript now reflect this: Compared with other methods of complete S-gene and whole genome sequencing, our techniques reduce the number of steps (and hence time) of library preparation. For example, the original ARTIC protocol used for whole genome sequencing includes expensive reagents such as NEBNext Ultra II
DNA Library Prep Kit for Illumina [https://www.protocols.io/view/covid-19-artic-v3-illumina-library-construction-an-j8nlke665l5r/v4] that lead to increase the cost and time resources. This fact means that researchers need to complete all steps including targeted amplification, DNA end repair, adenylation, ligation with adapter sequences, amplification indices, and purification using AMPure XP beads three times per protocol. Our methods allow avoiding additional steps due to primers containing Illumina adapters, which helps to reduce the total reagent cost by almost twice.

2. Only Delta samples are analyzed in the study. However, the authors highlighted that the described pipeline could quickly optimize the primers to fit emerging variants, and this declaration should be verified.

It should be noted that the time frame of our study was from winter 2021 to summer 2022. In the summer of 2021, Delta became the dominant variant across the territory of our country. Since the end of 2021, Omicron has dominated. Finally, now sublines BA.1-BA.5 are presented widely in our region. As a result, we examined all the samples gathered by our partners from hospitals during this time. We had to constantly adapt because the virus presented sublines changed. In particular, Figure 1 shows a rapid narrowing of the 95% confidence interval that is related to modifications in the primer set made with the help of our pipeline. The spread of the Delta strain largely caused the requirement to replace primers. Figures 1 and 3 thus serve as examples of how we can adjust to the
emergence of genetic changes in the viral genome. The text now contains subtle indications that the dominant strains have changed over the course of our study (lines 220-221 and 231-232 respectively):

Then, when new sublines spread, the 95%-confidence region gradually widens and the average numbers of sufficiently covered regions fall, indicating a degradation in resulting data quality.

Moreover, the spreading of the Delta variant in the summer of 2021 worsened the situation. Hopefully, it will no longer mislead a reader.

Also, we want to thank you personally for pointing out the minor issues in our paper. We fixed them in the updated version (lines 2, 9, 50, 68-69, 80, and Fig. 5).

Thanks to you, hopefully, we were able to improve our article.

Reviewer 2 Report

This paper mainly introduces two methods and strategies, which are used for S gene amplification and sequencing, and whole genome library construction and sequencing. By optimizing the experimental steps and automatic analysis and calculation, the detection time and cost have been shortened.

However, there are several typing errors in the abstract, e.g. 'peopl', 'spreading of the ?'

Abbreviations should be written in full when they firstly appeard, e.g. LAMP

What is the reduction of cost?

Author Response

Dear Reviewer,

First of all, we would like to express our deep gratitude for the evaluation of our manuscript and comments.

As for the reduction of the cost, due to variable prices for reagents in our country, we roughly estimated that sequencing of a single sample became 40-50% less expensive. Lines 378–388 of the updated version of our manuscript now reflect this: Compared with other methods of complete S-gene and whole genome sequencing, our techniques reduce the number of steps (and hence time) of library preparation. For example, the original ARTIC protocol used for whole genome sequencing includes expensive reagents such as NEBNext Ultra II
DNA Library Prep Kit for Illumina [https://www.protocols.io/view/covid-19-artic-v3-illumina-library-construction-an-j8nlke665l5r/v4] that lead to increase the cost and time resources. This fact means that researchers need to complete all steps including targeted amplification, DNA end repair, adenylation, ligation with adapter sequences, amplification indices, and purification using AMPure XP beads three times per protocol. Our methods allow avoiding additional steps due to primers containing Illumina adapters, which helps to reduce the total reagent cost by almost twice.

Also, we want to thank you personally for pointing out the minor issues in our paper. We fixed them in the updated version (lines 2, 9, 50, 68-69, 80, and Fig. 5).
We do hope were able to improve our article.